# ENO2 Promotes Colorectal Cancer Metastasis by Interacting with the LncRNA CYTOR and Activating YAP1-Induced EMT

**DOI:** 10.3390/cells11152363

**Published:** 2022-08-01

**Authors:** Chunwei Lv, Hongfei Yu, Keyi Wang, Chaoyi Chen, Jinlong Tang, Fengyan Han, Minglang Mai, Kehong Ye, Maode Lai, Honghe Zhang

**Affiliations:** 1Department of Pathology, Research Unit of Intelligence Classification of Tumor Pathology and Precision Therapy of Chinese Academy of Medical Sciences (2019RU042), Zhejiang University School of Medicine, Hangzhou 310058, China; lcw13284520839@126.com (C.L.); yuhongfei@zju.edu.cn (H.Y.); 11918046@zju.edu.cn (C.C.); fengyan999@zju.edu.cn (F.H.); 21918036@zju.edu.cn (M.M.); 12018134@zju.edu.cn (K.Y.); 2Key Laboratory of Disease Proteomics of Zhejiang Province, Zhejiang University School of Medicine, Hangzhou 310058, China; 3Central Laboratory, Affiliated Hangzhou First Peoples Hospital, Zhejiang University School of Medicine, Hangzhou 310006, China; wky_jacky@163.com; 4Department of Pathology, The Second Affiliated Hospital, Zhejiang University School of Medicine, Hangzhou 310009, China; 21018274@zju.edu.cn; 5Cancer Center, Zhejiang University, Hangzhou 310058, China; 6Department of Pharmacology, China Pharmaceutical University, Nanjing 210009, China

**Keywords:** ENO2, colorectal cancer, metastasis, CYTOR, YAP1

## Abstract

The glycolytic enzyme enolase 2 (ENO2) is dysregulated in many types of cancer. However, the roles and detailed molecular mechanism of ENO2 in colorectal cancer (CRC) metastasis remain unclear. Here, we performed a comprehensive analysis of ENO2 expression in 184 local CRC samples and samples from the TCGA and GEO databases and found that ENO2 upregulation in CRC samples was negatively associated with prognosis. By knocking down and overexpressing ENO2, we found that ENO2 promoted CRC cell migration and invasion, which is dependent on its interaction with the long noncoding RNA (lncRNA) CYTOR, but did not depend on glycolysis regulation. Furthermore, CYTOR mediated ENO2 binding to large tumor suppressor 1 (LATS1) and competitively inhibited the phosphorylation of Yes-associated protein 1 (YAP1), which ultimately triggered epithelial–mesenchymal transition (EMT). Collectively, these findings highlight the molecular mechanism of the ENO2–CYTOR interaction, and ENO2 could be considered a potential therapeutic target for CRC.

## 1. Introduction

Colorectal cancer (CRC) has a significant impact on health worldwide, ranking third in incidence and second in mortality [1]. Distant metastasis is the primary cause of CRC-associated mortality, and approximately one fourth of patients with advanced CRC have liver metastasis [2]. Therefore, investigating the regulatory mechanisms that control tumor cell metastasis, as well as exploring novel key biomarkers of CRC, could significantly improve its diagnosis and treatment [3].

Epithelial–mesenchymal transition (EMT) is an important process that leads to cancer cell metastasis [4], in which cells undergo a conversion from an epithelial to a mesenchymal phenotype, whereby cells develop loose cell–cell interactions and become motile [5]. Previous studies have demonstrated that numerous genes and signaling pathways can trigger the transcription of EMT-related genes and the onset of EMT in epithelial tumors. Our previous study demonstrated that the lncRNA CYTOR has a vital role in regulating EMT in CRC [6]. Moreover, we identified the glycolytic enzyme enolase 2 (ENO2) as a CYTOR-binding molecule, but its function and molecular mechanisms in CRC metastasis remain unclear.

ENO2 is a general tumor marker for neuroendocrine tumors [7] that has also been reported to correlate with tumor progression in some types of cancer [8,9]. ENO2 is an important member of the enolase family, which catalyzes the synthesis of phosphoenolpyruvate (PEP) to mediate glycolysis and gluconeogenesis [10]. Some studies have suggested that ENO2 promotes metastasis by activating and enhancing glycolysis [11,12,13], while others have attributed this promotion to non-glycolytic pathways [14]. Muller et al. first showed that inhibition of ENO2 by the homozygous deletion of ENO1 selectively suppressed the growth and tumorigenic potential of glioblastoma cells [15]. Then, a heterocyclic, cell-permeable inhibitor of enolase, which seemingly binds outside the active site, was described by Jung [16]. Some studies have shown that ENO2 has an important role in other cellular processes in addition to glycolysis [17,18] and is commonly referred to as the moonlight protein [19,20].

The transcription coactivator Yes-associated protein (YAP) determines several functions, such as organ growth, cell adhesion and EMT [21]. YAP1 is an important downstream effector of the Hippo pathway [22], which is regulated by a phosphorylation/dephosphorylation process [23]. Phosphorylation of YAP1 at Ser127 by large tumor suppressor kinase (LATS) results in cytoplasmic sequestration and ubiquitin-dependent degradation [24]. Earlier studies have shown that the activation of YAP1 is essential for CRC metastasis and that YAP1 overexpression could be an independent predictor of CRC prognosis [25]. Although several theories have been proposed [26], the corresponding mechanism has not been fully elucidated.

In this study, we demonstrated that ENO2 expression was correlated with metastasis, poor prognosis in CRC and increased migration and invasion as an oncogene. ENO2 plays pro-metastatic roles by activating YAP1-induced EMT, which is dependent on its interaction with lncRNA CYTOR.

## 2. Materials and Methods

### 2.1. Cell Lines and Reagents

The human CRC cell lines RKO, HT29, DLD1 and SW480 were purchased from the Shanghai Cell Collection (Shanghai, China) and cultured in RPMI 1640 (GIBCO, Shanghai, China). The human embryonic kidney cell line 293T was cultured in Dulbecco’s modified Eagle’s medium (GIBCO, Shanghai, China). Media were supplemented with 10% fetal bovine serum (FBS; HyClone, Shanghai, China). Plasmids and siRNAs (Gemma, Shanghai, China) were transfected into cell lines with LipoD293 (SignaGen, Rockville, MD, USA, cat# SL100668) and Genmute siRNA transfection reagent (SignaGen, Rockville, MD, USA, cat# SL100568), respectively. Further information is presented in Appendix A.

### 2.2. TCGA and Gene Expression Omnibus (GEO) Analysis

The gene expression profiles and clinical data of ENO2 for 430 patients were downloaded from the TCGA database (https://tcga-data.nci.nih.gov/tcga/ (accessed on 6 April 2020) and the GEO database (https://www.ncbi.nlm.nih.gov/geo/ (accessed on 6 April 2020). The GSE49355 and GSE14297 datasets contained 15 and 18 primary metastasis paired samples. Significant differences between the two groups were analyzed with GraphPad Prism 8.

### 2.3. Quantitative Real-Time PCR (qRT–PCR)

qRT–PCR was performed in a Roche 480II Real-Time PCR System. Total RNA was isolated using TRIzol reagent (Invitrogen, Shanghai, China, cat#15596018). cDNA was synthesized by HiScript II Reverse Transcriptase (Vazyme, Nanjing, China, cat# R223-01), and the qPCR reagent was SYBR qPCR Master Mix (Vazyme, Shanghai, China, cat# Q711-02). All primers used in this assay are listed in Appendix A.

### 2.4. Western Blotting and Immunoprecipitation

We performed Western blotting as described previously [27]. Protein samples (30 µg) were subjected to electrophoresis on 10% SDS–polyacrylamide gels (FUDE Biotech, cat#FD2060) and then transferred to nitrocellulose membranes (Millipore, Shanghai, China, cat# HATF00010). The membranes were probed with the appropriate antibodies. All antibodies used in this research are listed in Appendix A. The blots were washed 3 times with TBST and labeled with infrared fluorescence-conjugated goat anti-rabbit IgG or goat anti-mouse IgG. Then, secondary antibodies were used for near-infrared fluorescent detection, performed on an Odyssey Infrared Imaging System (LI-COR) according to the manufacturer’s instructions.

For the immunoprecipitation assay, approximately 1 × 10^7^ cells were lysed in 1 ml of IP lysis buffer (Beyotime, Shanghai, China, cat# P0013) containing protease inhibitors. Two micrograms of specific primary antibody and the corresponding IgG (as a negative control) were mixed with 20 µL of protein-A/G Agarose beads (Santa Cruz, Shanghai, China, cat# sc-2003) for 1 h at room temperature, followed by centrifugation at 2500× *g* for 10 min. The supernatant was removed, and 500 µL of the cell lysate was added to each tube for incubation overnight at 4 °C with slow mixing. The next day, the beads were washed 3 times with IP lysis buffer and centrifuged for 5 min at 2500× *g*. The complexes were eluted with 60 µL of 1× SDS–PAGE loading buffer and then boiled for 5 min. Then, 50 µg of protein was loaded for each lane for Western blotting to analyze the immunoprecipitation results.

### 2.5. Construction of the ENO2 and CYTOR Knockout (KO) Cell Lines

To construct ENO2KO cells, the pLentiCRISPRv2 vector containing a single-guide RNA (sgRNA) targeting ENO2 was transfected into RKO and SW480 cells to introduce a premature stop codon in the first exon of the ENO2 gene. To construct CYTORKO cells, the pLentiCRISPRv2 vector containing a sgRNA targeting the CYTOR promoter and a pUC57 vector containing a homologous recombination donor to delete the CYTOR promoter (−1461~+142) were transfected into RKO cells. Forty-eight hours after transfection, 4 µg/mL puromycin was added to the culture medium. After one week, puromycin-resistant single cells were seeded into a 96-well plate. These single-cell clones were expanded, and the KO effect was examined by PCR and RT–qPCR at the genomic and transcriptional levels. Clones without Cas9 editing were selected as mock cells. The sgRNA sequences and PCR primers are presented in Appendix A.

### 2.6. Construction of ENO2 Overexpression and Mutant Plasmids

The full-length or partial ENO2 CDS with a 3×FLAG tag was cloned into the overexpression vector pCDH-CMV-MCS-EF1-puro. The ENO2 mutants were designed and constructed by using the Mut Express II Fast Mutagenesis Kit V2 (Vazyme, Nanjing, China).

### 2.7. Cell Viability and Transwell Assays

A CCK8 kit (Boster, Wuhan, China, cat# AR1160) was used to determine relative cell viability. According to a previously described protocol [28], cells were cultured in 96-well plates and treated with CCK8 reagent for 1 h, and the medium was removed and placed into new 96-well plates. Then, the absorbance was measured at 450 nm using a microplate reader.

Transwell assays were performed using 24-well transwell chambers (Corning). Two hundred microliters of serum-free medium containing 5 × 10^4^~1 × 10^5^ cells was added to the upper chamber, and 800 µL of RPMI 1640 medium with 10% FBS was added to the lower chamber to act as the chemo-attractant. Following incubation for 24–48 h, cells migrating to the lower surface of the filter were fixed with 4% paraformaldehyde at room temperature for 15 min. Then, the cells were stained with 0.1% crystal violet solution for 5 min and images were captured using a microscope.

### 2.8. Metabolic Flux Analysis

Cellular bioenergetics were determined using a Seahorse XFe96 Analyzer (Agilent Seahorse Technologies, Santa Clara, CA, USA). Cells were seeded at a concentration of 1.5 × 10^4^ cells per well in RPMI 1640 medium. Before measurement, the cells were incubated with basal RPMI (Agilent Seahorse Technologies, Santa Clara, CA, USA, cat# 103576-100) without glucose or sodium pyruvate. Measurements were performed with a Seahorse XF Glycolysis Stress Test Kit (Agilent Seahorse Technologies, Santa Clara, CA, USA, cat# 103020-100) and the extracellular acidification rate (ECAR) was simultaneously recorded and calculated by Wave software for Seahorse XFe96.

### 2.9. Glucose Consumption, Lactate Production and Enolase Activity Assays

The concentrations of glucose and lactate in the culture medium were measured after 24, 48 and 72 h with a glucose test kit (Solarbio, Hangzhou, China, cat# BC2505) and a lactate assay kit (Jiancheng Bioengineering, Nanjing, China), respectively.

Enolase activity was measured via NADH oxidation in a pyruvate kinase–lactate dehydrogenase coupled assay, as previously described [29]. Cells were lysed in 20 mM Tris–HCl, 1 mM EDTA and 1 mM β-mercaptoethanol (pH 7.4), followed by sonication for 30 s 3 times. Enolase activity was recorded by measuring the oxidation of NADH at an excitation wavelength of 340 nm and an emission wavelength of 460 nm.

### 2.10. Cell Immunofluorescence (IF) and Immunohistochemistry (IHC)

For the IF assay, the sections were deparaffinized with xylene, hydrated with graded alcohol aqueous solutions and blocked using Blocking Buffer (Beyotime, Shanghai, China, cat# P0260) for one hour, followed by incubation with the corresponding primary antibody (ENO2, 1:50; YAP1, 1:200) overnight at 4 °C. A secondary antibody (Alexafluor R546, 1:400; Alexafluor M647, 1:1000) was then applied for an additional 1 h of incubation at room temperature in the dark. After washing with PBS, images were captured with an OLYMPUS confocal microscope imaging system.

For the IHC assay, tissue sections were treated with citrate buffer (pH 6.0) under high pressure for 5 min to complete antigen retrieval. Blocking was performed with 10% bovine serum for 30 min at room temperature. Then, the sections were incubated with a specific primary antibody overnight at 4 °C, followed by incubation with a secondary antibody (ZSGB-BIO, Beijing, China, cat# PV6001) at room temperature for 30 min. Staining was performed with 3,3′-diaminobenzidine (DAB) (ZSGB-BIO, cat# PV8000). Then, the sections were counterstained with hematoxylin, dehydrated and covered with a cover slip. Based on the immunohistochemical staining intensity and positive area, ENO2 expression was evaluated, and the IHC score was calculated by multiplying the positive staining area by the staining intensity. ENO2 expression values were classified into low (score = 0–1), middle (score = 2–3) and high levels (score ≥ 4).

### 2.11. RNA Immunoprecipitation (RIP)

The RNA immunoprecipitation assay was performed according to the manufacturer’s instructions using the RIP-Assay kit (Millipore, cat# 17-700).

### 2.12. RNA-Seq and Gene Set Enrichment Analysis (GSEA)

Total RNA was extracted, sequenced and analyzed by Bioacme (Wuhan, China). Differential gene expression analysis was accomplished with the Cuffdiff program in the Cufflinks package. Genes with fold change > 2.0 and *p* < 0.05 were defined as differentially expressed gene candidates for further analysis and qPCR validation. RNA-seq data were deposited (PRJNA824622).

The GSEA program was used to analyze gene expression data at the level of gene sets. The gene sets collection database MSigDBv6.2 C5 (GO gene sets) was used, and the gene sets with sizes of 15 to 500 were selected. Gene permutation was used to generate a null distribution, and all other parameters were the default selections.

### 2.13. Mouse Metastasis Model

All animal experiments were performed in accordance with the protocol approved by the Institutional Animal Care and Use Committee at Zhejiang University (Ethics Committee number: 19931). RKO mock and RKO ENO2-KO cells were infected with lentivirus containing pGKV5, which was co-transfected with the packaging plasmids pMD2G and pSPAX2, and then treated with4 µg/mL G418 to screen for the cells stably expressing luciferase. Then, two groups of cells were injected into the spleens of immunodeficient mice (NOD–SCID–gamma, male, 5 weeks old) by surgical operation. Each mouse was injected with 1 × 10^6^ cells suspended in 100 μL of PBS. The metastatic foci of the tumors were detected and observed with an IVIS Spectrum system and hematoxylin–eosin (HE) staining after 12 weeks.

### 2.14. Statistical Analysis

All data are shown as the mean and standard deviation (SD). The data were analyzed by Student’s *t* test and an X^2^ test using GraphPad Prism 8 software. Kaplan–Meier survival analysis was performed using IBM SPSS Statistics v23 software with the log-rank (Mantel–Cox) test. The results were considered statistically significant at *p* < 0.05.

## 3. Results

### 3.1. ENO2 Was Generally Highly Expressed in CRC Tissues and Negatively Correlated with Prognosis

First, we reanalyzed ENO2 expression in CRC samples from the TCGA database to determine the potential role of ENO2 in CRC progression (Appendix A). We found that high ENO2 expression was significantly associated with poor prognosis (*p* = 0.0019), which was consistent with a previous study by Pan et al. [30]. To determine whether the ENO2 level was correlated with tumor metastasis, we performed immunohistochemical staining of ENO2 in the 184 samples from CRC patients. We observed that ENO2 protein expression was higher in CRC tissues than in adjacent normal tissues (Figure 1A). We next analyzed the relationship between pathological features and ENO2 expression. As shown in Appendix A, ENO2 was correlated with the 5-year survival of CRC patients (*p* = 0.016), but there was no relationship between ENO2 expression and other clinical pathological parameters. These data indicated that the expression of ENO2 may play important roles in regulating the development of CRC. Then, survival analysis showed that high ENO2 expression was also significantly correlated with poor prognosis (*p* = 0.005) in local samples (Figure 1B), which was consistent with the TCGA dataset results. From the GEO repository, we retrieved two datasets containing paired primary and metastatic cancer samples from 33 individuals and compared the expression levels of ENO2. We found that ENO2 expression was significantly higher in metastatic tumors than in primary tumors (Figure 1C). These data suggested that ENO2 was upregulated in CRC tissues and was negatively correlated with prognosis.

### 3.2. ENO2 Promoted CRC Cells’ Migration and Invasion

To investigate the roles of ENO2 in CRC metastasis, we first detected endogenous ENO2 mRNA and protein expression in CRC cell lines, which showed a relatively higher level of ENO2 in RKO and SW480 cells than in DLD1 and HT29 cells (Figure 2A). Then, we knocked down ENO2 in RKO and SW480 cells with specific small interfering RNAs (siRNAs). The transwell assay showed that the ENO2 knockdown significantly inhibited cell migration and invasion (Figure 2B and Appendix A). Then, overexpression of ENO2 in DLD1 cells significantly promoted CRC cell migration and invasion (Figure 2C). More interestingly, with the re-expression of ENO2 in ENO2 knockout cells, the potential of migration and invasion was rescued (Figure 2D and Appendix A). To validate the roles of ENO2 in vivo, we performed a murine model of hepatic metastases via spleen injection of RKO ENO2 knockout cells. The result showed that ENO2 knockout cells formed less metastatic foci in the liver than mock cells (Figure 2E). Thus, our findings demonstrated that ENO2 functioned as a pro-metastatic agent in CRC.

### 3.3. ENO2 Promoted Metastasis Independently of Altering the Glycolytic Rate in CRC

Some studies have reported that ENO2 promotes migration and invasion by regulating the glycolytic pathway in other types of cancer [13,31]. Therefore, we knocked down ENO2 in CRC cells (Figure 3A) and investigated cell proliferation, the consumption of glucose and the accumulation of lactic acid. Unexpectedly, knockdown of ENO2 neither suppressed cell proliferation (Figure 3B) nor reduced the accumulation of lactate and consumption of glucose in the medium (Figure 3C,D). To further verify the lack of effect of ENO2 on glycolysis in CRC, a metabolic flux assay was employed to determine the ECAR of CRC cells after ENO2 knockdown. We selected PKM2 as a positive control, which has been reported to be the key rate-limiting glycolytic enzyme in CRC [32]. The results showed that PKM2 knockdown caused a significant decrease in the maximum ECAR in RKO cells, but ENO2 knockdown did not (Figure 3E). In addition, we detected the total enolase activity in ENO2-silenced RKO cells and found that the absence of ENO2 did not affect the total enolase activity (Figure 3F). The same results were observed in SW480 cells (Figure 3G,H). Then, overexpression of ENO2 in DLD1 cells did not affect the maximum ECAR or the total enolase activity (Figure 3I,J). To further exclude the glycolytic role of ENO2 to promote CRC metastasis, ENO2 with mutations in the substrate binding site (Mut-SBD) was overexpressed (Figure 3K). As expected, ENO2 Mut-SBD still significantly promoted the migration and invasion of CRC cells (Figure 3L). Together, these data indicated that ENO2 promoted CRC cell migration and invasion without altering the glycolytic rate.

### 3.4. ENO2 Induced EMT by Interacting with the lncRNA CYTOR

EMT has been considered an important molecular mechanism of CRC metastasis [33,34,35]. To determine whether ENO2 promotes migration and invasion in CRC cells by regulating EMT, we assessed the changes in expression of several important markers of EMT. The results showed that ENO2 knockdown in CRC cells resulted in increased expression of E-cadherin and occludin (OCLN) and decreased expression of N-cadherin and SLUG at the protein levels (Figure 4A,B). Overexpression of ENO2 in DLD1 and ENO2-silenced SW480 cells caused a decrease in the expression of epithelial markers and an increase in the expression of mesenchymal markers (Figure 4 C and Appendix A).

Our previous study demonstrated that CYTOR could promote metastasis by inducing EMT in CRC [6]. The RNA pull-down assay combined with mass spectrometry (MS) identified ENO2 as a binding protein of CYTOR (Figure 4D). Here, we further validated the specific interaction between CYTOR and ENO2 through a RIP assay, which showed that ENO2 could bind to CYTOR but not to the two other control lncRNAs, MIR4435-2HG and LINC01133 (Figure 4E,F and Appendix A). However, it remains unknown whether ENO2 plays pro-metastatic roles after it interacts with CYTOR. Therefore, we detected the migration and invasion potential when ENO2 was knocked down in CYTORKO cells Figure 4G and Appendix A). The results showed that the decrease in CYTOR levels significantly inhibited migration and invasion, but knocking down ENO2 did not further inhibit the migration and invasion potential of RKO cells (Figure 4H). Similar results were observed in SW480 cells (Appendix A). To clarify the functional domain of ENO2, we evaluated the functions of a series of ENO2 mutants in ENO2-silenced RKO cells (Figure 4I). The results showed that the domain composed of amino acids 374–380 of ENO2 was the key for the ENO2-mediated promotion of CRC metastasis (Figure 4J). According to the different properties of different amino acids, we further constructed point mutation ENO2 expression vectors. Transwell and RIP assays in ENO2KO cells (Figure 4K and Appendix A) showed that mutation of proline at residue 380 impaired the potential of ENO2 to promote metastasis and interact with CYTOR (Figure 4L). These results demonstrated that ENO2 induced EMT by interacting with CYTOR in CRC and that the proline residue at position 380 was a key functional site of ENO2.

### 3.5. ENO2 Activated YAP1 by Suppressing YAP1 Phosphorylation

To further investigate the molecular mechanism by which ENO2 promotes EMT, we performed RNA-seq to analyze the differentially expressed genes between ENO2KO CRC cells and mock cells. GSEA showed that the Hippo signaling pathway was significantly negatively regulated by ENO2 (Figure 5A). However, as a major effector of the Hippo signaling pathway, YAP1 expression was not decreased at the mRNA level. Therefore, we detected the protein expression and phosphorylation level of YAP1 at Ser127. Our results revealed that the phosphorylation of YAP1 at Ser127 was significantly increased, while the total protein level of YAP1 was repressed in the absence of ENO2 (Figure 5B–D). These results were further confirmed by IF (Figure 5E,F). Next, we knocked down YAP1 to verify its oncogenic effect in CRC cells. The decreased expression of YAP1 significantly inhibited migration and invasion and decreased the expression of ENO2-related EMT markers (Figure 5G,H and Appendix A). Additionally, ENO2 knockdown in CRC cells with low expression of YAP1 did not further suppress migration and invasion (Figure 5I,J and Appendix A). Moreover, YAP1 knockdown completely reversed the increased migration and invasion potential induced by ENO2 in DLD1 cells (Figure 5K). Collectively, we found that ENO2 inhibited the phosphorylation of YAP1 and led to the oncogenic accumulation of total YAP1, which triggered the EMT process in CRC cells.

### 3.6. CYTOR Mediated the Interaction between ENO2 and LATS1 to Inhibit the Degradation of YAP1

We next investigated the underlying mechanism by which ENO2 inhibits the phosphorylation of YAP1. LATS1 phosphorylates YAP1 at the Ser127 site, resulting in the inactivation and degradation of YAP1 [26,36]. Thus, coimmunoprecipitation (Co-IP) experiments were performed to determine the interaction between ENO2, YAP1 and LATS1 (Figure 6A). The results showed that LATS1 could bind to YAP1 and ENO2. Moreover, endogenous RIP showed that LATS1 could also bind to CYTOR (Figure 6B). CYTOR can modulate multiple signaling pathways in cancer cells [37]. To determine whether CYTOR regulates the interaction between ENO2, LATS1 and YAP1, we detected the interaction of LATS1 with YAP1 and ENO2 in CYTOR knockout RKO cells by Co-IP. As expected, decreased expression of CYTOR significantly enhanced the binding between LATS1 and YAP1 and significantly reduced the interaction between LATS1 and ENO2 (Figure 6C). Then, we performed a similar assay in ENO2-silenced SW480 cells, which also showed that ENO2 and YAP1 competitively bind to LATS1 (Figure 6D–F). Taken together, these data show that CYTOR mediates the interaction between ENO2 and LATS1 and that ENO2 knockout reduced the expression of YAP1 and inhibited the metastasis of CRC cells (Figure 6G).

## 4. Discussion

Until now, most ENO2-related studies have focused on the clinical applications of ENO2 using retrospective analysis, and only a few publications have discussed the specific mechanism by which ENO2 affects tumorigenesis and development at the cellular and molecular levels. In recent years, the tumor-promoting and metastatic effects of ENO2 in cancer have gradually attracted the attention of cancer researchers, but its biological mechanism has not been fully elucidated. Energy metabolism in tumor cells is heavily dependent on glycolysis. Therefore, some tumor cells show large increases in the expression and activity of glycolysis-related enzymes to gain a competitive advantage in a glucose-limited environment. Initially, we attributed the pro-metastatic effect of ENO2 to its promotion of the glycolytic rate. However, we found that changes in ENO2 expression did not affect the rate of glycolysis in CRC cells. In mammals, enolase has three isoforms, α-enolase, β-enolase and γ-enolase, encoded by ENO1, ENO3 and ENO2, respectively. ENO1 is the major isoform of enolase in cancer cells, accounting for 75–90% of the cellular enolase activity [29]. In a series of studies by Muller, ENO2-specific inhibitors did not significantly affect enolase activity in ENO1 wild-type cells [38,39]. All the above information suggests that ENO2 promotes CRC metastasis through non-glycolytic pathways. By detecting the effect of ENO2 on the expression of EMT markers in CRC cells, we observed that ENO2 was positively correlated with the protein levels of several typical mesenchymal phenotype markers, such as SLUG and N-cadherin, and negatively correlated with the protein levels of the epithelial phenotype markers OCLN and E-cadherin.

Interestingly, we found that ENO2 might be considered an RNA-binding protein that can bind to the lncRNA CYTOR. Furthermore, the key functional region of ENO2 shows a large amount of overlap with the region that binds to CYTOR, as determined by constructing a series of ENO2 truncation mutants. More intriguingly, we discovered an association between ENO2 and the Hippo pathway through RNA-seq and pathway enrichment analysis. The Hippo pathway not only participates in the regulation of organ size, stem cell function and tissue regeneration but also plays an important role in the occurrence and development of tumors [40,41]. As described in previous studies, the interaction of CYTOR with sam68 and NCL occurs mainly in the nucleus. While ENO2 is widely regarded as a plasma protein, the interaction between CYTOR and ENO2 is also located in the cytoplasm. Therefore, we believe that the new mechanism discovered in this study is independent of previous findings. Although we have proven that ENO2 could bind to LATS1 and inhibit the binding of LATS1 to YAP1, further study is needed to verify whether ENO2 competitively occupies the binding site of LATS1 and YAP1 or whether ENO2 binds to LATS1 to cause conformational changes that block its interaction with YAP1. In addition, the poor tumorigenicity of RKO in the mice spleen–liver model resulted in the sample size in each group being insufficient, thus requiring further investigation. Therefore, all of the above issues will be explored in follow-up studies.

In conclusion, our study confirmed the pro-metastatic function of ENO2 in CRC and clarified its non-glycolytic-related molecular pro-metastatic mechanism. The present study demonstrated novel regulation patterns of ENO2 dependent on its interaction with the lncRNA CYTOR in CRC metastasis, which not only improves our current understanding of the function of CYTOR during the development of CRC but also provides a novel biomarker of or therapeutic target for CRC metastasis.

## Figures and Tables

**Figure 1 cells-11-02363-f001:**
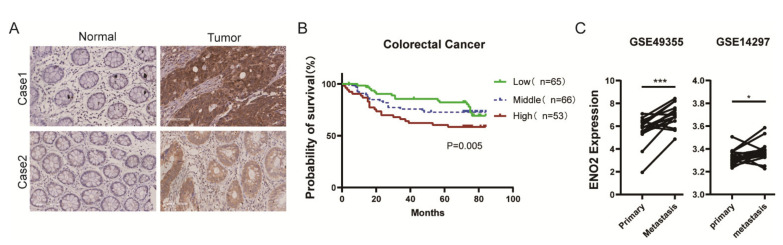
ENO2 was upregulated in CRC tissues and negatively correlated with poor prognosis. (**A**): Representative immunohistochemical staining of the ENO2 protein in CRC samples and in the corresponding adjacent normal tissue. (**B**): Overall survival analysis in 184 CRC patients with low (*n*  =  66), middle (*n* = 65) or high (*n*  =  53) levels of ENO2 by Kaplan–Meier analysis. (**C**): GEO analysis (GSE49355 and GSE14297) (http://www.ncbi.nlm.nih.gov/geo/ (accessed on 6 April 2020). * *p*  <  0.05, *** *p*  <  0.001.

**Figure 2 cells-11-02363-f002:**
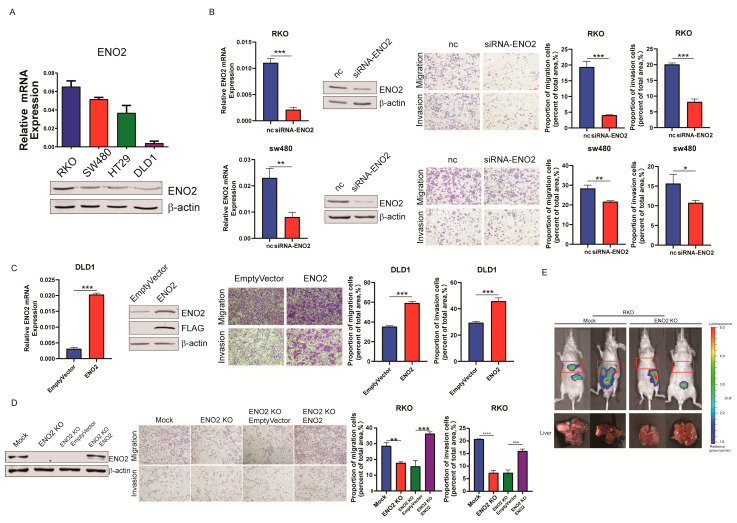
ENO2 promoted CRC migration and invasion in vitro and metastasis in vivo. (**A**): Expression of ENO2 RNA (up) and protein (down) in CRC cell lines normalized to β-actin. (**B**): Identification of ENO2 knockdown and its effects on RKO and SW480 migration and invasion. Shown from left to right are mRNA as measured by qPCR, protein levels as measured by Western blot, representative transwell images and statistical results from the transwell migration and invasion assays. (**C**): The migration and invasion of DLD1 cells overexpressing ENO2. (**D**): Representative transwell results of the phenotypic rescue of ENO2KO cells. ENO2 was reintroduced into ENO2-silenced RKO cells. Characterization of ENO2 re-expression and the corresponding statistical analysis of migration and invasion. (**E**): Metastatic foci in a mouse model of liver metastasis. The red rectangles indicate the mouse liver region, *n* = 2. Values are presented as the means  ±  SDs, * *p*  <  0.05, ** *p*  <  0.01, *** *p*  <  0.001, **** *p* < 0.0001.

**Figure 3 cells-11-02363-f003:**
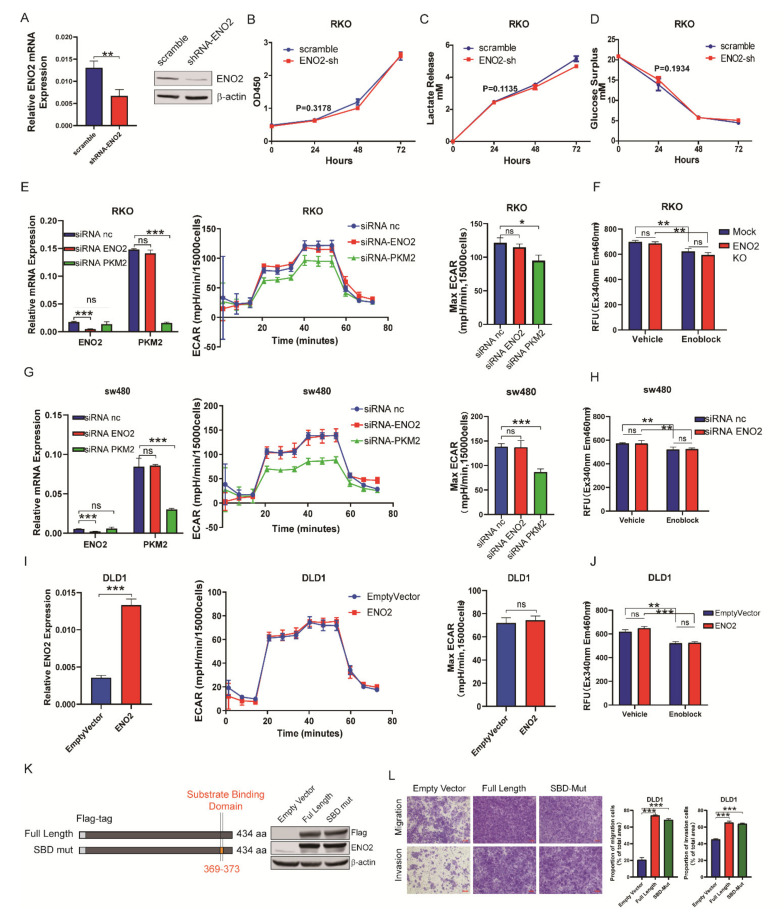
ENO2 promoted metastasis independently of altering the glycolytic rate in CRC. (**A**): qPCR and Western blot analysis confirmed ENO2 knockdown in RKO cells. (**B**–**D**): ENO2 knockdown did not significantly inhibit cell proliferation, glucose consumption or lactate production in RKO cells. (**E**): The ECAR of the indicated cells was detected by using a Seahorse XFe96 Extracellular Flux Analyzer, and the maximum glycolytic rates are summarized. PKM2 was knocked down as a positive control. *n*  =  3 per group. (**F**): The total enolase activity of RKO ENO2 knockdown cells was detected by performing a relative fluorescence unit (RFU) assay. Enoblock was used as a positive control. (**G**–**J**): Seahorse and total enolase activity of SW480 ENO2 knockdown cells and DLD1 ENO2 overexpressing cells. (**K**): Characterization of ENO2 with mutations in the substrate binding region. (**L**): Mutations in the substrate binding region of ENO2 did not suppress its metastatic effect. Data are shown as the mean ± SD. * *p* < 0.05, ** *p* < 0.01, *** *p* < 0.001. ns: non significance.

**Figure 4 cells-11-02363-f004:**
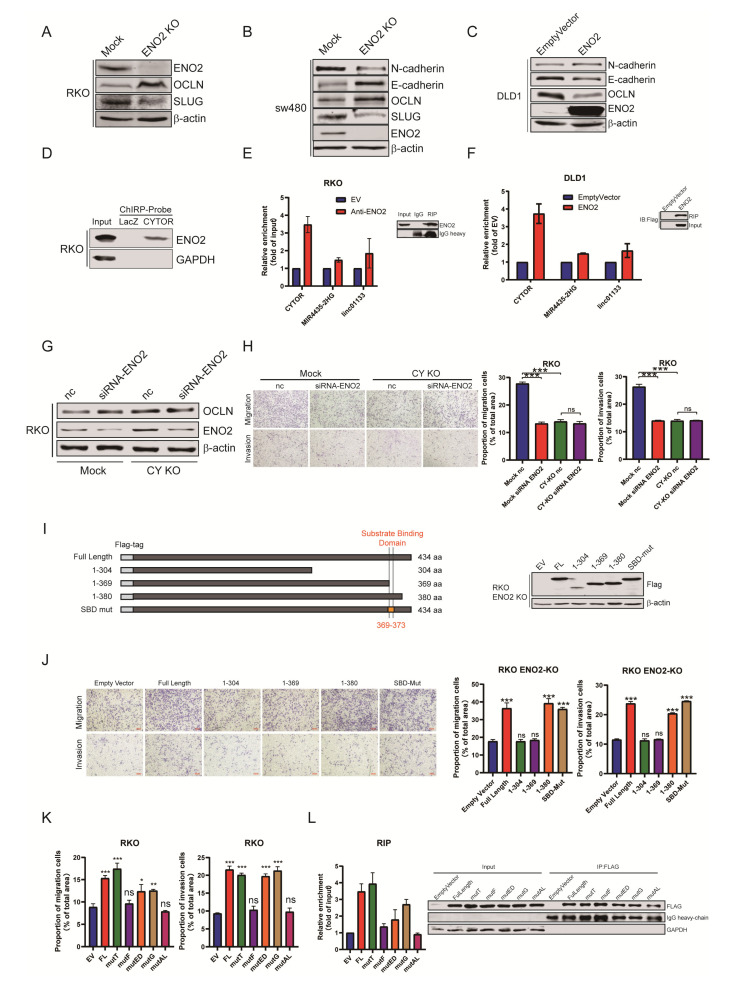
ENO2 induced EMT through a lncRNA CYTOR-dependent pathway. (**A**–**C**): ENO2 induced the expression of mesenchymal genes. Western blotting was used to analyze the expression of EMT-related genes in stable cell lines. (**D**): RNA pull-down assay results showing that ENO2 was a CYTOR-binding protein. (**E**,**F**): The RIP experiment detected the binding of ENO2 to CYTOR and several EMT-related lncRNAs. (**G**): The OCLN level in RKO cells with ENO2 knockdown and CYTOR KO. (**H**): The migration and invasion of RKO cells with ENO2 knockdown and CYTOR KO. (**I**): Identification of full-length ENO2 and different truncated ENO2 sequences reintroduced into RKO ENO2 KO cells. (**J**): The functions of the ENO2 truncation mutants depicted in (**I**). (**K**): The function of single point mutations to amino acid residues 374–380 of ENO2. (**L**): RIP assay detected the binding of ENO2 single point mutation to CYTOR. Data are shown as the mean ± SD. * *p* < 0.05, ** *p* < 0.01, *** *p* < 0.001. ns: non significance.

**Figure 5 cells-11-02363-f005:**
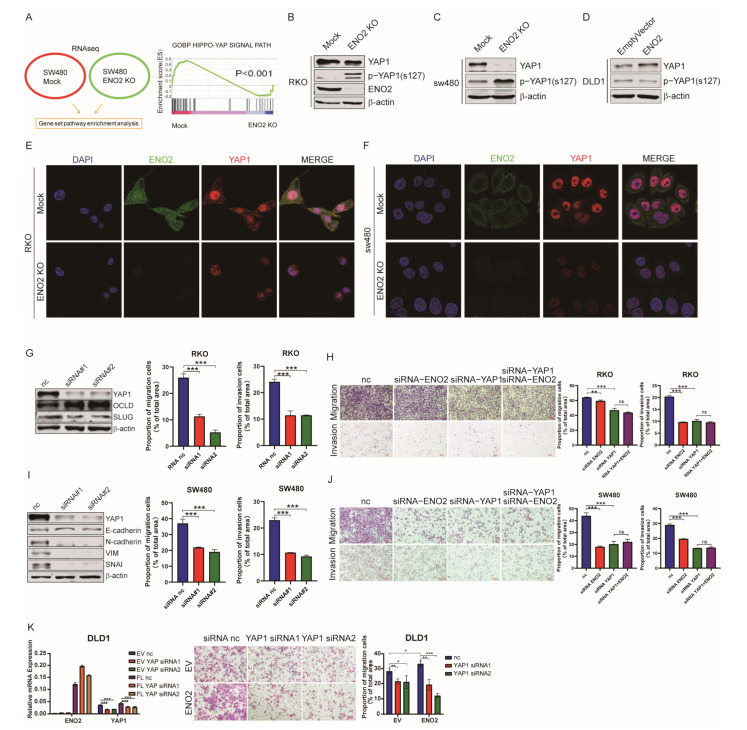
ENO2 induced YAP1 activation by suppressing its phosphorylation. (**A**): ENO2 stimulates the Hippo pathway in SW480 cells. (**B**–**D**): The phosphorylation of YAP1 Ser127 and the degradation of YAP1 were detected by Western blot after ENO2 knockdown. (**E**,**F**): IF assay showing ENO2 and YAP1 expression. Scale bar = 10 μm. YAP1 is stained red, ENO2 is stained green, and the nucleus is stained blue (DAPI). (**G**,**H**): Identification of YAP1 and ENO2 knockdown in RKO cells (**G**). Migration and invasion of RKO cells with YAP1 and ENO2 knockdown (**H**). (**I**,**J**): Identification of YAP1 and ENO2 knockdown in SW480 cells (**I**). Migration and invasion of SW480 cells with YAP1 and ENO2 knockdown (**J**). (**K**): The migration of DLD1 cells overexpressing ENO2 with YAP1 knockdown. Data are shown as the mean ± SD. * *p* < 0.05, ** *p* < 0.01, *** *p* < 0.001. ns: non significance.

**Figure 6 cells-11-02363-f006:**
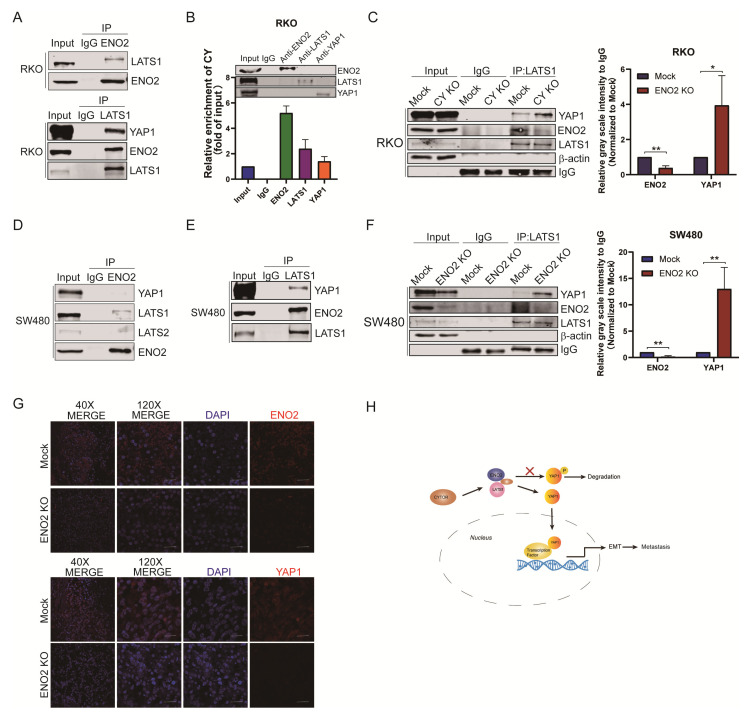
CYTOR mediated ENO2 binding to LATS1 and inhibited the degradation of YAP1. (**A**): Co-IP assay of anti-LATS1 in RKO cells. (**B**): RIP assay of CYTOR with anti-ENO2, anti-LATS1 and anti-YAP1 in RKO cells. **(C**): Co-IP assay of anti-LATS1 in RKO mock/RKO CYTOR KO cells. (**D**,**E**): Co-IP assay of anti-ENO2 (**left**) and anti-LATS1 (**right**) in SW480 cells. (**F**): KO of ENO2 significantly enhanced the binding between LATS1 and YAP1. (**G**): Immunohistochemical staining revealed the differences in ENO2 and YAP1 expression in implanted orthotopic splenic tumors from the two groups (magnification, 200×). (**H**): Schematic representation of the ENO2 signal transduction pathway in CRC. Data are shown as the mean ± SD. * *p* < 0.05, ** *p* < 0.01.

## Data Availability

RNA-seq data were deposited (PRJNA824622).

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
