# Peer review of "ENO2 Promotes Colorectal Cancer Metastasis by Interacting with the LncRNA CYTOR and Activating YAP1-Induced EMT"

_cells, 2022, doi:10.3390/cells11152363_

Round 1

Reviewer 1 Report

In this manuscript by Zhang and colleagues, the authors investigate the pro-metastatic function of glycolytic enzyme, enolase 2 (ENO2) in colorectal cancer (CRC) pathogenesis.  ENO2 belongs to the enolase family of proteins which catalyses the conversion of 2-phosphoglycerate to phosphoenolpyruvate. In this manuscript authors show that ENO2 is upregulated in CRC tissues and demonstrates higher expression in metastatic tissue compared to primary tumors with a negative correlation with patient prognosis. Using loss and gain of function approach, ENO2 was shown to promote CRC cell migration and invasion in vitro. Further, ENO2 knockout cell line showed lesser liver metastasis in vivo compared to control cells.  This pro-metastatic effect of ENO2 was independent of its catalytic function in glycolysis since neither ENO2 knock down or over expression influenced glucose consumption or extracellular acidification rate. Mechanistically, ENO2 was shown to promote the expression of EMT markers and suppress epithelial marker genes via its interaction with cytoskeleton regulator RNA (CYTOR) through a key proline residue (pro380) in ENO2.  Using NGS analysis, the authors profiled global gene expression in ENO2KO CRC cells and found Hippo signaling pathway to be affected. Functionally, ENO2 inhibits YAP1 inactivation (by phosphorylation) since ENO2 KO cells showed increased phospho YAP1. Moreover, ENO2 overexpression induced migration and invasion could be reversed by YAP1 knockdown suggesting that ENO2 exerts its pro-metastatic function via stabilizing YAP1. Through RNA IP studies, authors demonstrate that CYTOR (which is known to bind to ENO2) also binds to LATS1, an upstream negative regulator of YAP1.  It is revealed that CYTOR promotes the interaction of ENO2 with LATS1 thereby inhibiting its interaction with YAP1 and thus stabilizing YAP1.  

The authors have performed a thorough investigation in this manuscript in determining the pro-metastatic function of ENO2. All experiments have been conducted with appropriate controls and multiple approaches (siRNA/shRNA/knockout) and model systems (in vitro, in vivo tumorigenesis, metastasis) have been used to prove the hypothesis. This manuscript can be accepted for publication with the following minor comments addressed.   

Minor comments:

1.     Sequencing data for to show premature stop codon introduction in ENO2 KO cells need to be provided

2.     Data for genomic deletion of CYTOR is also lacking in the manuscript (either via PCR or sanger sequencing)

3.     In section 2.4, the authors should provide the amount of protein (micrograms) used for immunoprecipitation rather than the volume of lysate used.

4.     In their previous manuscript, the authors show that CYTOR also interacts with NCL and Sam68 proteins. Could the interaction between ENO2/LATS with CYTOR and its interaction with NCL/Sam68 be mutually exclusive? This could be discussed.

5.     Figure S1E: The last panel in Fig S1, should be labled as siRNA-ENO2 + siRNA CY

6.     In several sections, the number representing the power of ten is not superscripted. (For instance, 5X 104 instead of 5X 104 cells)

7.     No immunofluorescence protocol for cells is given in the methods section.

Reviewer 2 Report

In this article, Lv et al. identified a glycolysis-independent mechanism of ENOS promoting the metastasis of colorectal cancer. The authors showed that ENOS could interact with LATS1 in a lncRNA-CYTOR-dependent manner and trigger EMT by competitively inhibiting the phosphorylation of YAP1. This article is interesting and comprehensive, which provides a novel clue for the development of CRC clinical therapy. However, there are a few questions that should be addressed.

1.     The authors did not use the words “knockdown” and “knockout” correctly (lines 371, 377, etc.), which will make the readers confused.

2.     In line 377, the existing evidence in this article is not strong enough to support that ENO2 can alter the nuclear localization of YAP1.

3.     To avoid the off-target effect of siRNA, the authors should use at least 2 siRNAs for the knockdown of a specific gene.

Reviewer 3 Report

The dysregulation of glycolytic enzyme enolase 2 (ENO2) is a major contributor to cancers. However, the potential molecular mechanisms of ENO2 in colorectal cancer (CRC) metastasis are unclear. In this study, the authors performed a comprehensive analysis to study the expressions of ENO2 in CRC and established the molecular mechanism of the ENO2-CYTOR interaction. Overall, the study is well conducted, and the conclusions are well supported by the data. I have a few questions listed below. 

1.   It is not clearly stated in the method section about the number of gene expression profiles downloaded from TCGA database. And the GEO IDs should be provided for a replication purpose.

2. Some words are accidently placed together, for example, ‘andnegatively’ in the subtitle of section 3.1.  
